



# Impact of the Guinea Coast upwelling on atmospheric dynamics, precipitation and pollutant transport over Southern West Africa

Gaëlle de Coëtlogon[1], Adrien Deroubaix[2,4], Cyrille Flamant[1], Laurent Menut[2], and Marco Gaetani[3]

[1]Laboratoire Atmosphères, Observations Spatiales, IPSL, Sorbonne Université, Paris, 75252, France
[2]Laboratoire de Météorologie Dynamique, Ecole Polytechnique, IPSL, Ecole Normale Supérieure, Université Paris-Saclay, Sorbonne Université, CNRS, Palaiseau, 91128, France
[3]Scuola Universitaria Superiore IUSS, Pavia, 27100, Italy
[4]IUP, Institute of Environmental Physics, University of Bremen, Bremen, D-28359, Germany

**Correspondence:** Gaëlle de Coëtlogon (gdc@latmos.ipsl.fr)

**Abstract.** In West Africa, the zonal band of precipitation is generally located around the southern coast in June before migrating northward towards the Sahel in late June / early July. This gives way to a relative dry season for coastal regions from Ivory Coast to Benin called "little dry season" which lasts until September - October. Previous studies have noted that the coastal rainfall cessation in early July seems to coincide with the emergence of an upwelling along the Guinea Coast: the aim of this study is to

investigate the mechanisms by which this upwelling impacts the precipitation, using a set of numerical simulations performed with the regional atmospheric model Weather Research and Forecasting (WRF v 3.7.1). Sensitivity experiments highlight the response of the atmospheric circulation to an intensification, or conversely a reduction, of the strength of the coastal upwelling: they clearly show that the coastal upwelling emergence is responsible for the cessation of coastal precipitation by weakening the northward humidity transport, thus decreasing the coastal convergence of the humidity transport and inhibiting the deep

atmospheric convection. In addition, the diurnal cycle of the low-level circulation plays a critical role: the land breeze controls the seaward convergence of diurnal anomaly of humidity transport, explaining the late night / early morning peak observed in coastal precipitation. The emergence of the coastal upwelling strongly attenuates this peak because of a reduced land-sea temperature gradient in the night and a weaker land breeze. The impact on the inland transport of anthropogenic pollution is also shown with numerical simulations of aerosols using the CHIMERE chemistry-transport model: warmer (colder) SST

increase (decrease) the inland transport of pollutants, especially during the night, suggesting an influence of the upwelling intensity on the coastal low-level jet. The mechanisms described have important consequences for inland humidity transport and the predictability of the West African Monsoon precipitation in summer.

*Keywords:* West African Monsoon, Guinea Coast, coastal precipitation, ocean influence, seasonal cycle, diurnal cycle, coastal upwelling

# 1   Introduction

Precipitation in West Africa is mostly controlled by a monsoon driven by a very strong meridional temperature and humidity contrasts between the eastern Tropical Atlantic and the dry continent further north (Parker et al., 2017). In response to the



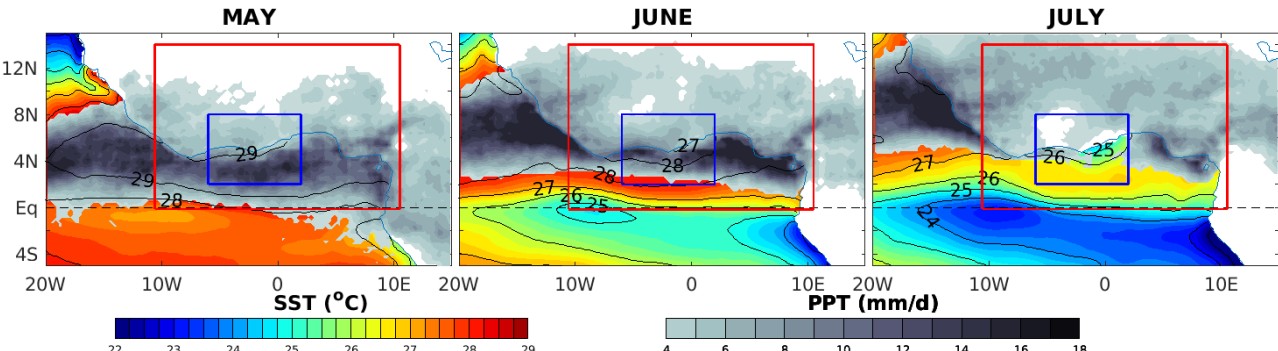

**Figure 1.** *Observations from May (left), June (center) and July (right) 2008-2017: sea surface temperature (SST, black contours with 1°C intervals and red-green-blue shading) and Tropical Rainfall Measuring Mission (TRMM) precipitation exceeding 4 mm/d (gray shading). The red and blue frames show the simulation coverage and the sub-region under scrutiny 6°W-2°E / 2°N-8°N.*

annual cycle of the insolation, precipitation exhibits a strong seasonal latitudinal cycle (Hagos and Cook 2007, Thorncroft et al. 2011, Maranan et al. 2018). The precipitation maximum is located just south of the Guinea Coast (i.e. the southern coast

of West Africa, located along about 5°N) between late April and early July (Figure 1, left and middle). It shifts northward in late June / early July until September, bringing precipitation to the Sahel region and leaving the Guinea Coast relatively dry, during the so-called "little dry season" (Figure 1, right). The Guinea Coast experiences a second rainfall peak in October – November when the monsoon recedes (not shown), as precipitation is mainly localized over the ocean during the rest of the year (Sultan and Janicot 2003, Zhang and Cook 2014).

Along the coastline, the major rainy season around June is called "Guinea Coast rainfall" (or GCR, Nguyen et al. 2011). Its onset is controlled by the emergence of the eastern Atlantic equatorial upwelling (also called "cold tongue") in late May (Leduc-Leballeur et al. 2013, Meynadier et al. 2015). Its cessation in July generally coincides with the development of coastal upwelling along the Guinea Coast from Ivory Coast to Benin, where the sea surface temperature (SST) decreases from about 30°C in May to 24-25°C in August (Odekunle and Eludoyin 2008, Ali et al. 2011, Kouadio et al. 2013). At interannual

timescales, Ali et al. (2011) found strong correlations between the GCR variability and both equatorial and coastal upwellings, and Bakun (1978) observed an important rainfall reduction along the coast where the coastal upwelling is the strongest (Figure 1, right). Other studies suggested an influence of the Guinea Coast upwelling on nearby continental precipitation (Gu and Adler 2004, Kouadio et al. 2013, Nnamchi and Li 2011). Nevertheless, Tanguy et al. (2022) were the first to propose a mechanism explaining the coastal upwelling impact on GCR. Using satellite observations (of convective clouds mainly) and

ERA5 reanalyses between 2008 and 2015, they suggested that the emergence of the Guinea Coast upwelling weakens the low-level Southerlies just before they meet the mainland, which, in turn, decreases coastal convergence and thus inhibits deep convection and precipitation. However, their results were based on composite analyses based upon an empirically determined date **of the apparition of the coastal upwelling**. Therefore, numerical simulations forced by coastal upwellings of varying intensities are necessary to assess unambiguously the mechanisms involved.





The Dynamics Aerosol Cloud Chemistry Interactions in West Africa (DACCIWA, Knippertz et al. 2015, Knippertz et al. 2017) project offers a database to analyze coastal upwelling, through observations and modelling. Numerical simulations of meteorology and atmospheric composition were carried out with the WRF-CHIMERE modelling system for the period June 1st and July 8th, 2016 (Deroubaix et al. 2019, Menut et al. 2019). In the present work, the role played by the coastal upwelling on the transition between GCR in June and the beginning of the little dry season in early July is investigated with

similar numerical simulations. We seek to quantify the impact of the coastal upwelling on regional atmospheric conditions and anthropogenic pollution transport. We also want to estimate how far inland this influence is propagated by the monsoon transports of humidity and pollutants from the coastal areas northward. Two ensemble of sensitivity simulations, forced either by a weaker upwelling (resulting in warmer coastal SST) or a more intense one (resulting in colder coastal SST), are therefore tested and compared to an ensemble of reference simulations, and the impact of SST anomalies on surface winds, surface wind

divergence, precipitation, humidity transport and pollutant transport is analysed. The production of the reference ensemble simulation is discussed in section 2. The design of the weaker or stronger upwelling ensemble simulation is described in section 3. The response of atmospheric dynamics and precipitation to changes in coastal SST is examined in section 4. The impact of coastal SST on the transport of anthropogenic pollution is studied in section 5. A summary and discussion concludes this work in section 6.

**2   Reference ensemble simulation (RefES)**

The modelling system couples the Weather Research and Forecasting (WRF) meteorological model (v 3.7.1, Powers et al. 2017) and the CHIMERE chemistry-transport model (v 2017, Mailler et al. 2017). We use a regional domain (10 km × 10 km) extending from $1^o$S to $14^o$N and from $11^o$W to $11^o$E (Figure 2), with 32 vertical levels (up to 50 hPa) which are projected on 20 levels for CHIMERE (up to 300 hPa) without chemistry but with passive tracers, as it has been done in Deroubaix et al. (2019)

from which we use the same simulation set-up. The main simulation was conducted with this configuration from June 1st to July 8th, 2016, when the surface ocean temperature drops the fastest in the coastal upwelling region (from about $28^o$C to $25^oC$). The National Oceanic and Atmospheric Administration National Centers for Environmental Prediction (NOAA NCEP) real-time global SST (Thiébaux et al. 2003) is used to force the SST in the WRF-CHIMERE simulations.

    In order to identify significant changes in coastal atmospheric dynamics resulting from a SST modification in the sensitivity

experiments described in the next sections, nine additional simulations were produced, each slightly modifying the SST forcing fields: in the second simulation, the SST on June 1 at 0000 UTC was replaced by the SST of 0600 UTC; in the third run, the SST at 0600 UTC was replaced by the SST of 1200 UTC; and so forth... These minor modifications do not modify significantly the SST field that forces the reference simulations before June 3rd, but they are relevant in order to produce an ensemble of 10 independent simulations. Since they are all forced by a similar SST field after June 3rd, their spread can be considered as an

indication of the physical and numerical variability of the model.

    In the following, we present the average of the 10 simulations computed over their last two weeks (i.e. June 25th – July 8th 2016), considering the previous period as time for the model to spin up before the coastal SST becomes and remains colder than





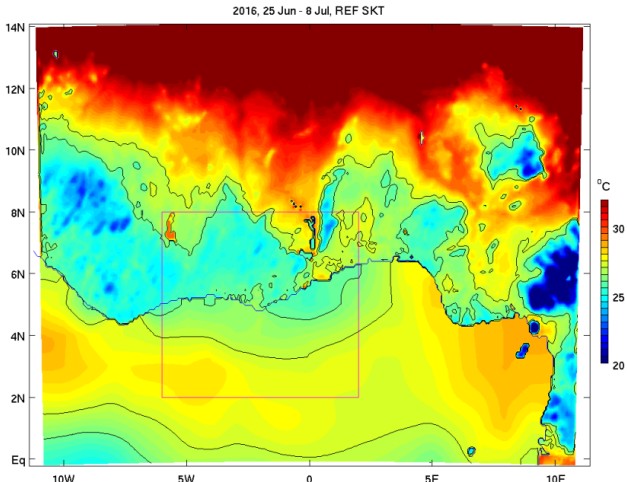

**Figure 2.** Jun 25th to Jul 8th, 2016: skin temperature (SKT in $^o$C, shading) from the Reference ensemble simulation (RefES). The red frame highlights the sub-region of $6^oW$-$2^oE$/$2^oN$-$8^oN$ shown in Figure 7 and Figure 10. The black contours represent the isotherms of $26^oC$ and $27^oC$.

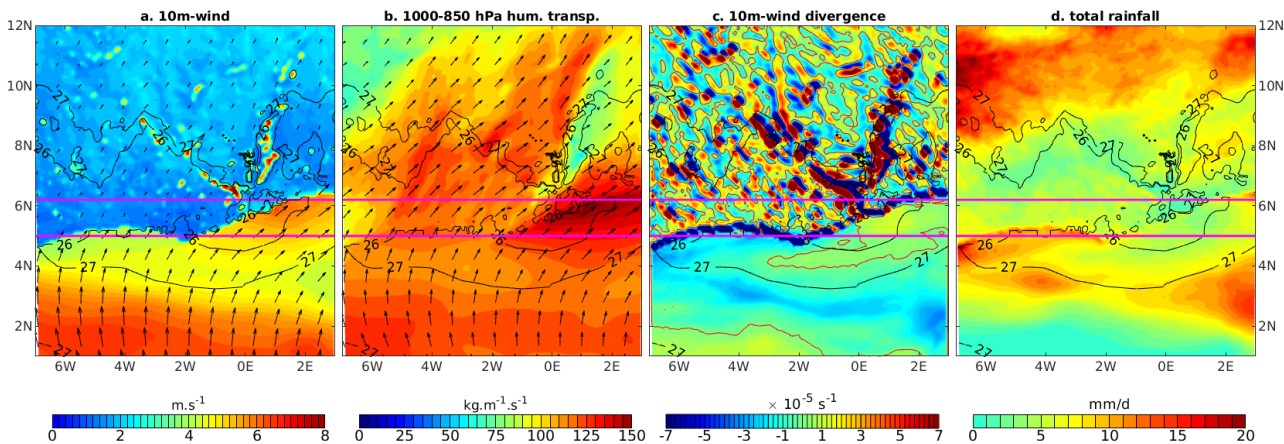

**Figure 3.** Jun 25th to Jul 8th, 2016: reference ensemble simulation (RefES). a. 10m-wind speed (shading) and direction (arrows). b. magnitude (shading) and direction (arrows) of humidity transport integrated between 1000 hPa and 850 hPa. c. 10m-wind divergence (shading, zero in red contour). d. precipitation (shading, mm/h). The black contours represent the isotherms of $26^oC$ and $27^oC$. Horizontal pink lines indicate the latitudinal extend of the coastline in the simulation domain.

$27^oC$. It will be referred to as the "reference ensemble simulation", or RefES thereafter. RefES is then presented for spatial and vertical averages of key meteorological variables over the studied period in the coastal upwelling region. It is also the RefES
that will be used with the CHIMERE model to analyze the impact of the transport on aerosol concentrations.



Figure 3 shows skin temperature (thereafter SKT), 10-m wind speed, 10-m wind divergence, total precipitation, and humidity transport integrated between 1000 hPa and 850 hPa (i.e. within the monsoon flow, see below). Near-surface southerlies strongly weaken when encountering the continent (Figure 3a) because surface roughness and heat fluxes are much larger over land than over sea. As a result, strong convergence is found at 10 m all along the coastline (Figure 3c). Over the continent, the divergence pattern is dominated by dipoles of 10-m divergence / convergence associated with the topographic features in the Lake Volta region, namely the Mampong range (south of $8^oN$, ranging from $2^oW$ to $0^oE$) and the Akwapim-Togo range around $1^oE$ (ranging from $6^oN$ to $11^oN$).

A weaker belt of surface wind convergence (of about -2 or -3 $\times 10^{-5}s^{-1}$) is also found over the ocean, around $2^oN$-$3^oN$ in the east and just south of the $27^oC$ SST contour further west, resulting from a slowing down of southerlies over SST colder than $27^oC$ that surrounds the coastal upwelling (Figure 3a, c). Indeed, surface wind speed tends to decrease when blowing over colder SST: a first mechanism to explain this is a reduction in the vertical flux of horizontal momentum, due to an increase in vertical stability above the sea surface, preventing the surface wind from mixing with the - generally stronger - upper boundary layer wind (Sweet et al. 1981, Wallace et al. 1989, Hayes et al. 1989). The second mechanism is an adjustment of the pressure gradient to the SST gradient via hydrostatic balance (Lindzen and Nigam 1987). In addition, east of $2^oW$, the wind strengthens and turns east along the Togo-Benin coastline, probably because of a positive downwind SST gradient in the northeastward direction east of $0^oE$, a zonal land-sea gradient, and the shape of the coast (see also Figure 11 and 12 in Flamant et al. 2018).

Over the ocean, the strong link between precipitation and surface wind convergence is well known: between 65% and 90% of global precipitation are associated with low-level lines of convergence (Weller et al., 2017). Hence, precipitation maximum of up to 0.5 mm/h fits the convergence belt just south of the $27^oC$ isotherm, whereas a minimum of 0.2 mm/h is collocated with the weakly divergent surface winds near the coast, east of $1^oW$ (Figure 3c-d). Indeed, since the marine atmospheric boundary layer is nearly saturated in humidity, surface wind convergence often corresponds to convergence of humidity transport, which favours convection and precipitation (Meynadier et al., 2015).

Over the continent, despite much weaker surface winds, humidity transport in the monsoon flow is at least as strong as over the ocean except east of Lake Volta (Figure 3b), because the near-surface atmosphere contains more water vapour over the continent than over the ocean at constant pressure level (see discussion below). This results in a divergence of humidity transport over the continent near the coast (not shown), which may explain the lower precipitation rates seen between the coast and $10^oN$ (Figure 3d).

Figure 4 shows the vertical section of the atmosphere up to 500 hPa averaged between $6^oW$ and $2^oE$, where the coastline is located between roughly $5^oN$ and $6.2^oN$ as framed in pink lines (horizontal lines in Figure 3, vertical lines in Figure 4). SKT and precipitation latitudinal profiles averaged between $6^oW$ and $2^oE$ are also plotted for comparison (Figure 4c, d). Two important features of the West African Monsoon are clearly seen in the zonal velocity: the near-surface West African Westerly Jet (Pu and Cook 2010) between $5^oN$ and $10^oN$, and the African Easterly Jet (AEJ) around 650 hPa north of $8^oN$ (Figure 4a). It also shows that the largest southerly winds are confined under 850-900 hPa, and that they lie below a widespread convective circulation over the ocean evidenced by ascending motion (Figure 4a).



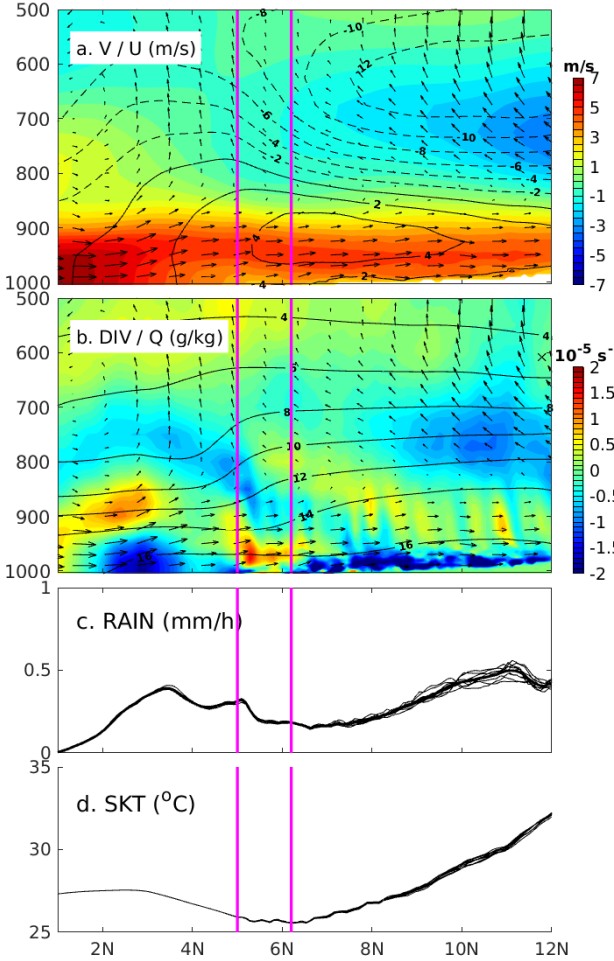

**Figure 4.** Jun 25th to Jul 8th, 2016, RefES, within the region 6°W-2°E: a. meridional velocity (shading) and zonal velocity (black contours, plain for positive, dashed for negative, intervals of 1 m.s$^{-1}$, null velocity in heavy black), b. horizontal divergence (shading) and specific humidity (black contours, g/kg), c. reference simulations (black profiles, average in thick line), and d. surface temperature profiles (black profiles, average in thick line). The vertical pink lines frame the coastline between 6°W and 2°E.

115    Near the ocean surface, two zones of convergence (i.e. negative divergence) are found (Figure 4b): one occurring as a response to the negative SKT gradient on the edge of the coastal upwelling between 2 and 3°N (already mentioned, see Figure 3c), and one associated with the coastal convergence at 5°N. These two convergence area correspond to peaks in precipitation (Figure 4d). Divergence is observed above each convergence area over the ocean, capped with a third layer of convergence, at the top or just above the monsoon flow, probably in response to local mass conservation. This "sandwich" is confined under

120    800 hPa in the coastal region and extends up to 700 hPa over the ocean (Figure 4b).





The SKT latitudinal profile exhibits a minimum down to 26$^o$C at the coast, and a SST increase of 3-4$^o$C due to the coastal upwelling within 100-200 km southward from the coast, to about 3$^o$N (Figure 4d). Unlike oceanic SKT, which is very close to the forcing SST and therefore identical in the ten reference simulations between June 25th and July 8th, the continental SKT is controlled by the surface layer scheme in the model which leads to some variance in the different runs composing RefES (black profiles). While SKT latitudinal profiles are weakly dispersed (Figure 4d), the dispersion of precipitation is important north of 7$^o$N (Figure 4c): the monsoon flow is indeed capped by a strong wind shear related to the presence of the AEJ at these latitudes, which is known to favor baroclinic instability and storm formation (Parker et al. 2017). Baroclinic instability enhances the sensitivity of the model to turbulence, and hence the dispersion between the simulations at these latitudes.

In summary, RefES reproduces well the transition period of the monsoon system from the GCR to the Sahel rainy season: precipitation is still present over the ocean, and the coastal upwelling is already well developed, contributing to trigger and maintain the belt of converging near-surface winds surrounding the 27$^o$C surface isotherm between 2$^o$N and 4$^o$N (Figure 3c, Figure 4b) and precipitation (Figure 3d, Figure 4c). This configuration is therefore appropriate for conducting sensitivity experiments with different upwelling intensities, and investigate the response of the atmospheric circulation to these changes.

## 3   Sensitivity experiment design

In order to build the SST forcing fields for the sensitivity experiments, the SST pattern which forces RefES (i.e. the reference SST) is modified in a way that the coastal upwelling is either dampened (weaker upwelling meaning warmer coastal SST, thereafter "warm ensemble simulation", or WarmES) or amplified (stronger upwelling meaning colder coastal SST, thereafter "cold ensemble simulation" or ColdES).

The methodology used to modify the SST is as follows:

i) a linear trend is first computed by least-squares fit between June 5th and July 31st, 2016 in the reference SST time serie at each oceanic grid point: large negative values emphasize the area where the coastal upwelling develops (Figure 5, top). Note that these trends were computed after June 5th (and until July 31st, when the upwelling is maximum) because the SST experiences a strong warming episode just before, as seen for the two particular points A and B (Figure 5, bottom, black).

ii) a threshold value of -0.052$^o C$/day is chosen in order to delimit the region where the SST is to be modified (Figure 5, top, black contour). This value fits the 27$^o$C SST contour, i.e. approximately the edge of the coastal upwelling (Figure 3a). Out of this area, SST timeseries remain unchanged.

iii) inside this area where the SST is modified, at each timestep and grid point, the difference between the threshold (-0.052 °C/day) and the local linear trend value is added to the SST time serie, thereby leading to a warmer SST time serie. All time series in the upwelling area thus have a linear trend capped at -0.052°C/day.

For example, the WarmES timeseries at points A and B show a progressively growing warm anomaly of about 0.5°C on June 25th and 1°C on July 7th, while the synoptic variability is conserved (Figure 5 A and B, red vs black time series). Note that the warm anomaly is slightly larger at point A than at point B, since the negative seasonal trend is also more pronounced in A than in B. Because the reference SST decreases by about 3 degrees between June 5th and July 8th at these locations, the magnitude





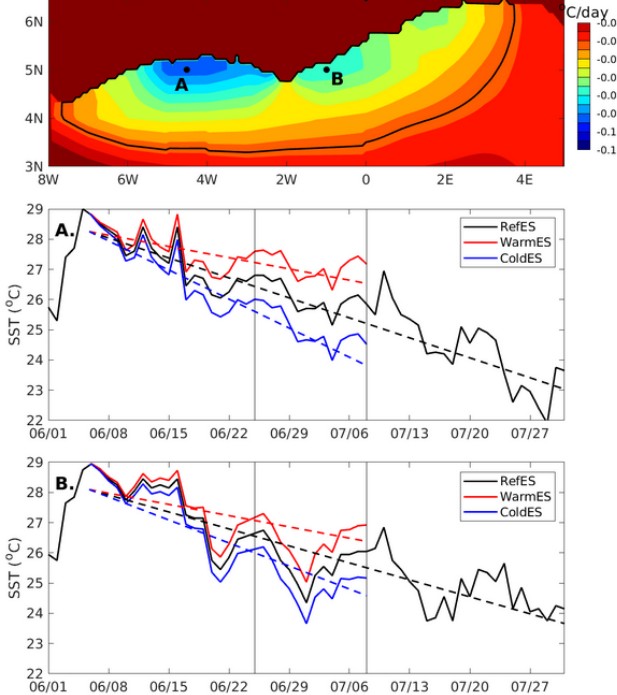

**Figure 5.** Top: linear trend in the RefES SST time series from Jun 5th to Jul 31st, 2016 (shading, $^oC$/day, with a threshold value of -0.052 $^oC$/day shown as black contour). Bottom panels for points A and B: RefSST time serie (solid black) and linear trend computed from June 5th onwards (dashed black), WarmES SST time serie (red) and linear trend (dashed red), and ColdES SST time serie (blue) and linear trend (dashed blue). Black vertical lines frame the period under scrutiny (June 25th – July 8th).

of the coastal upwelling is therefore dampened by about one third in WarmES. The resulting WarmES SST anomaly, averaged over the last two weeks of the simulation, exceeds 1$^o$C off Ivory Coast and is approximately 0.7$^o$C off Ghana (Figure 6a, black contours).

ColdES SST was built with similar SST anomalies than WarmES, but with the opposite sign (cold anomalies instead of warm): ColdES is then forced by a SST with cold anomalies, thereby simulating a coastal upwelling with a magnitude enhanced by about one third (Figure 5A and B, blue). The anomalies averaged over June 25th – July 8th in ColdES are then identical to those in WarmES, but cold instead of warm (Figure 6e).

Finally, the same methodology as for RefES was used to produce each of the 10 WarmES or ColdES simulations (i.e. with slight changes in SST in the first 2-3 days of the simulation). All the simulations were then averaged between June 25th and July 8th, 2016: for a given variable, the difference between the ensemble values in RefES and the ensemble values in WarmES (resp. ColdES) is considered significant when it passes a two-sample t-test at the 5% confidence level (i.e. with a p-value less than 5%).







**Figure 6.** June 25th to July 8th, 2016: differences between WarmES and RefES simulations (left) and between ColdES and RefES simulations (right). a.,e.: 10m-wind (speed in shading, m.s$^{-1}$) and surface temperature (black contours, intervals of 0.1$^o$C). b.,f.: divergence of 10m-wind (shading, s$^{-1}$). c., g.: total precipitation (mm/h). d., h.: humidity transport integrated between 1000 and 850 hPa (kg.m$^{-1}$.s$^{-1}$). Only difference values passing a two sampled t-test at the 5 % confidence level are shaded.

## 4   Atmospheric responses in WarmES and ColdES

The most striking effect of SST anomalies can be observed in the 10-m wind field: surface winds are stronger over warm SST anomalies (Figure 6a), and weaker over cold SST anomalies (Figure 6e). As already mentioned before, both vertical stability and pressure gradient mechanisms concur to reduce (strengthen) wind speed over cold (warm) SST regions (de Coëtlogon et al.

170   2014).

Off Ivory Coast, the maximum wind speed response to positive SST anomalies larger than 1$^o$C is up to +0.4-0.5 m.s$^{-1}$ in WarmES, whereas the response to negative SST anomalies of the same amplitude is on the order -0.7 - -0.8 m.s$^{-1}$ in ColdES.



This means that the response of surface winds to a SST anomaly is not symmetric for cold or warm anomalies: the weakening of surface winds over the cold surface is more pronounced than the strengthening over the warm surface. It may be explained by the fact that vertical stabilization effect on the wind over a cooling SST occurs only below a threshold of about 26$^o$C (Meynadier et al. 2015, see in particular their Fig. 8), since the SST is more often colder than 26$^o$C in ColdES than in WarmES in the coastal upwelling (Figure 5A-B).

In response to these changes in surface wind, the thin band of coastal convergence is amplified, as a consequence of stronger winds, in the warm case (Figure 6b, negative values indicate increase in convergence) and dampened by weaker winds in the cold case (Figure 6f, positive values indicate decrease in convergence). As explained in section 2, precipitation tends to follow the convergence pattern in the marine boundary layer, therefore more coastal precipitation is found in WarmES (Figure 6c) and less in ColdES (Figure 6g) along the Guinea Coast. The amplitude and offshore extension of precipitation anomalies are larger west of Cape Three Points than further east, reflecting the SST anomalies amplitude. In the warm case, maximal precipitation anomalies off Ivory Coast can exceed +0.2 mm/h, but are less than -0.2 mm/h in the cold case: contrary to surface wind, precipitation is more strongly impacted in the warm case than in the cold case. This signal is more robust west of Cape Three Points than further east, off Ghana and Togo, where anomalies are just above the significance threshold.

When the intensity of the upwelling is dampened (WarmES), humidity transport is enhanced from the ocean to the continent by up to 5 kg.m$^{-1}$.s$^{-1}$, due to a more intense monsoon flow, especially west of Cape Three Points (Figure 6d). When the upwelling is enhanced (ColdES), the humidity transport anomaly is on the contrary southwestward over and off Ivory Coast and Ghana, indicating a reduced northward humidity transport in the monsoon flow by about 5 kg.m$^{-1}$.s$^{-1}$ (Figure 6h). Note that these changes in the humidity transport magnitude are small compared to its mean value (about 100-150 kg.m$^{-1}$.s$^{-1}$, Figure 3b), but they are nonetheless significant, except east of Cape Three Points where there are only significant changes in the direction of the transport, which tends to be oriented more north / northwestward in WarmES and more southeastward in ColdES.

The impact of the coastal upwelling is further investigated in the low and middle troposphere up to 500 hPa with vertical meridional sections computed between 6$^o$W and 2$^o$E. The mean SKT and precipitation differences WarmES-RefES and ColdES-RefES are plotted (Figure 7c-d and d-h, heavy blue), as well as ten profiles of difference arbitrarily chosen (among 100 possible time series of WarmES-RefES and ColdES-RefES differences) in order to illustrate the typical dispersion in the numerical simulations (thin black). Even though modest (with amplitudes of slightly more than 0.5$^o$C, Figure 7d-h), SKT anomalies in the coastal upwelling region correspond to significant changes in precipitation within 100 km of the coast, with an increase by up to 0.1 mm/h in WarmES (Figure 7c) and a decrease by slightly less than 0.1 mm/h in ColdES (Figure 7g): these changes represent about one third of the mean rainfall at this latitude (i.e. 0.2-0.3 mm/h, Figure 4c), emphasizing how much the coastal precipitation is sensitive to the coastal SST. North of 6-7$^o$N, the precipitation anomalies are too small to clearly show a response emerging above the noise. Further analysis in the following therefore only focuses on the link between SKT and precipitation south of 6$^o$N.

In response to the warm SKT anomaly in WarmES, a negative pressure anomaly is found in the near-surface boundary layer between 3$^o$N and 5$^o$N (Figure 7a, black contours). The induced wind acceleration (Figure 7a, shading) clearly leads to





**Figure 7.** June 25th to July 8th, 2016: differences between WarmES and RefES (left column) and between ColdES and RefES (right column) within the region $6^oW-2^oE$. a.,e.: meridional-vertical circulation (black arrows, meridional velocity (shading) and geopotential height anomalies (black contours, dashed for negative, plain for positive, intervals of 0.2 m). e., f.: meridional-vertical circulation (black arrows), horizontal divergence (shading) and specific humidity (black contours, plain for positive, dashed for negative, intervals of 0.4 g/kg). c.,g.: precipitation (blue, mm/h). d.,h.: SKT (blue, $^oC$). In c.-h., 10 time series are shown (black) out of 100 possible time series from WarmES minus RefES or ColdES minus RefES, roughly indicating their dispersion. The coast latitude is framed by vertical pink lines. Only difference values of meridional velocity and horizontal divergence that pass a two sampled t-test at the 5 % confidence level are shown as shaded areas.





an increase in convergence below 900 hPa on the southern edge of the coast (Figure 7b, shading), explaining an increase in low-level humidity (Figure 7b, black contours) and coastal precipitation (Figure 7c). Convergence at the surface is capped

by divergence between 900 and 825 hPa, and a convergent area located slightly further north and up to 700 hPa, indicating an overall acceleration of south-westerlies below 800 hPa and observed as far north as $8^o$N (Figure 7b). In ColdES, opposite anomalies are found: the cold SKT anomaly creates a positive near-surface pressure gradient anomaly, forcing southward surface wind anomalies (Figure 7e). The resulting divergent anomaly (Figure 7f) decreases humidity under 850 hPa and inhibits the convection around $5^o$N, explaining the decreased coastal rainfall (Figure 7g).

In short, the emergence of the Guinea Coast upwelling in early July clearly induces a low-level atmospheric pressure gradient anomaly which weakens the near-coastal oceanic southerlies. It strongly dampens the coastal convergence within the monsoon flow, thereby inhibiting coastal convection and decreasing local precipitation, as well as inland humidity transport, particularly over Ivory Coast and Ghana west of Cape Three Points. This mechanism was already proposed by Tanguy et al. (2022) using observations and ERA5 reanalyses, but they also suggested an impact of coastal upwelling on the diurnal time scale via the

land-see breeze mechanism, which is also found in RefES: in the night and early morning, when the land (around $22^o$C, Figure 8a) is colder than the sea (around $27^o$C), a southward pressure gradient force weakens the strong southerlies that meet the continent, resulting in a low-level convergence (and thus increased precipitation) along and just south of the coast (Figure 8b), with divergence further north. During the afternoon and evening, when the land (up to $30^o$C) is warmer than the ocean (still around $27^o$C), this anomaly reverses: convergence occurs on the land side and favors precipitation over the continent, while

the latter is inhibited on the sea side via a low-level divergence (Figure 8b). Note that the wind is always from the south, because the amplitude of the diurnal anomalies (of about 0.5 m s$^{-1}$, Figure 8a) is only about 10% of the average wind (4 to 6 m s$^{-1}$, Figure 3a); however, these anomalies are directed southward and precipitation occurs over the ocean during the night and early morning, and turn progressively northward until the end of the day with precipitation over the continent in the late afternoon. In addition to dampening low-level coastal convergence at the seasonal timescale, a more intense coastal upwelling

has therefore a stronger impact during the night, since it reduces the land-sea surface temperature gradient during the night and early morning: the land breeze strength decreases, as well as low-level convergence and precipitation on the oceanic side of the coast (Figure 8d). Reducing the coastal upwelling strength leads to the exact opposite situation (Figure 8c).

   In summary, the emergence of the coastal upwelling leads to a decrease in coastal precipitation. We quantified this impact using numerical experiments: for a coastal precipitation of 0.3-0.4 mm/h on average in late June/early July (Figure 4c), a

dampened (increased) coastal upwelling of about 0.5°C (Figure 6d, h) results in an increase (decrease) in coastal precipitation of about 0.1 mm/h (Figure 6c, g). In addition, an examination of the diurnal cycle shows that most of the coastal precipitation occurs in the late night/early morning, with a peak of 0.5 mm/h around 0600 UTC (Figure 7b), due to the land breeze as discussed previously: the dampening (increase) of the upwelling leads to an increase (decrease) in morning precipitation of 0.1-0.2 mm/h (Figure 7c,d), following the reduction in the land-sea temperature gradient during the night, thus accounting for

much of the observed 0.1 mm/h reduction in daily mean precipitation.





**Figure 8.** June 25th to July 8th, 2016: time-latitude diagram (diurnal cycle) within the region $6^{o}$W-$2^{o}$E. In RefES: a. diurnal cycle of SKT (shading) and diurnal anomalies of 10-m wind (arrows), b. diurnal anomalies of 10-m wind divergence (shading) and diurnal cycle of precipitation (black contours, intervals of 0.1 mm/h). Differences between WarmES (c.) or ColdES (d.) and RefES of diurnal cycles of precipitation (shading) and 10-m surface wind (arrows). Only precipitation and velocity anomalies passing a two sampled t-test at the 5 % confidence level are plotted.





## 5  Pollutant transport

In previous sections, we have analysed the changes of atmospheric circulation due to a modification of the upwelling-related SSTs. Sensitive experiments show that wind fields are modified, thus impacting convection and precipitation patterns. Conse-
quently, we expect that the transport of pollutants emitted from the highly urbanized Guinea Coast are also impacted by coastal
upwelling SST-related modifications.

To look at the impact of coastal SST anomalies on the transport of pollutants, we use passive tracers released in the CHIMERE model from five major coastal cities, namely Abidjan (5$^o$18'N/4$^o$00'W, Ivory Coast), Accra (5$^o$33'N/0$^o$11'W, Ghana), Lomé (6$^o$08'N/1$^o$12'E, Togo), Cotonou (6$^o$21'N/2$^o$25'E, Benin) and Lagos (6$^o$27'N/3$^o$23'E, Nigeria), with constant tracers emission rates proportional to the city population. This approach allows to analyse pollutant transport modification that
are only due to atmospheric circulation because tracer emission is prescribed to be constant in time (see (Deroubaix et al., 2019) for details). It also allows to tag the emissions from a given city and compute its contribution to the overall tracer budget on each of the model grid points. We quantify the contribution at one of the DACCIWA super-site in Savè (Benin). The pollutant simulations are performed for the three main ensembles, RefES, ColdES and WarmES, in order to extract as clearly as possible the differences between the SST-induced atmospheric conditions.

We present the difference of tracer concentrations (in arbitrary units, a.u., summing the contribution of the five cities) averaged over the period from June 25th to July 8th between WarmES and RefES (Figure 9a) and between ColdES and RefES (Figure 9b). In both cases, the anomalies of coastal SST have an impact on the transport of tracers downstream of the five cities at the coast. A significant impact is also seen as far north as 8.5$^o$N for all cities. The concentration anomaly fields associated with the plumes emitted from individual cities (i.e. the juxtaposition of positive and negative anomalies) is less clear for Abidjan
than that from the other cities, which is related to the more complex dynamics of the monsoon flow to the west of Cape Three Points, possibly due to the fact that the near surface monsoon winds are almost parallel to the coastline east of 2$^o$W, as the monsoon winds are veering eastward inland (see Figure 3b for RefES).

When the upwelling is dampened (WarmES), the tracer concentration anomaly field over the continent exhibits a distinct structure downstream of the four easternmost cities (east of Cape Three Point) with positive anomalies to the northeast and
negative anomalies to the southeast (Figure 9a) which may be related to a counter-clockwise veering of the monsoon winds im-posed by a change in the coastal SST in the upwelling area. It is also worth noting that the transport of pollution is significantly enhanced over the ocean east of Lomé along the coast of Togo, Benin and Nigeria, as opposed to the tracers emitted from Abidjan. This feature, taking the form of a narrow band parallel to the coastline, is likely associated with the amplification of the land breeze (Figure 7). Downstream of Accra, the transport of urban pollutant tracers is seen to be enhanced close to Accra
(in connection with the land breeze) but decreased further away possibly due to the influence of the shape of the coastline.

When the upwelling is enhanced, in ColdES, the structure of the tracer concentration anomaly field over the continent is opposite to that seen in WarmES, thereby suggesting a clockwise veering of the monsoon winds imposed by the increase in the coastal SST (Figure 9b). East of Cape Three Point, there is almost no significant tracer concentration anomalies offshore, except downstream of Accra, suggesting that the land breeze circulation is weaker than in RefES when the upwelling is strengthened,




**Figure 9.** Upper panels: Difference maps of urban tracer transport modelled with (a) WarmES - RefES and with (b) ColdES - RefES using a constant release of urban tracers from five major coastal cities of the Gulf of Guinea (black dots): Abidjan (Ivory Coast), Accra (Ghana), Lomé (Togo), Cotonou (Benin) and Lagos (Nigeria), taking into account the city population (see (Deroubaix et al., 2019) for details). Lower panels: Hourly diurnal cycle of the urban tracer concentrations at the DACCIWA super site of Savè in Benin (green dot on upper panels) averaged over the period from June 25th to July 8th modelled with (c) WarmES - RefES and with (d) ColdES – RefES. Note that the green colors associated with Abidjan are not visible because the pollution of Abidjan (studied by the tracers emitted in Abidjan) does not reach the city of Savé for any of the three ensembles.

which is consistent with Tanguy et al. (2022). Overall, by comparing WarmES and RefES, the picture emerges that when the coastal upwelling is reinforced, the transport of urban pollutant tracers inland is reduced.

Even though the mean near-surface wind direction over Southern West Africa was shown to be quite constant during the period of simulation, there exists a marked variability of the strength of the monsoon flow at the diurnal scale in this region, as discussed in the previous section. This daily variability has an impact on the diurnal cycle of the cloud cover(Schuster





et al., 2013). It also has an impact on the inland transport of humidity from the ocean (Deetz et al. 2018, Adler et al. 2019) as well as pollutant from coastal cities (Deroubaix et al., 2019).

There are three noticeable phases during the day:

(i) from 0900 to 1500 UTC, there is an accumulation of pollutants at the coast due to convection and pollutants are not transported far from the emission sources. This period is defined as the "daytime drying" period;

(ii) from 1600 to 0200 UTC, there is the formation of the Low Level Jet. Pollutants accumulated along the coastline are transported below 500 m above ground level, within the boundary layer, toward the northeast, this period being defined as the "Atlantic inflow" period;

(iii) from 0300 to 0800 UTC, the intensity of the Low Level Jet decreases, thereby decreasing the northward transport of moisture and pollutants, this period being defined as the "moist morning" period.

CHIMERE-derived tracer simulations performed by Deroubaix et al. (2019) for the period July 1-7, 2016 have highlighted that the emissions from the above mentioned large coastal cities are likely to affect the air quality in remote cities located inland over 200 km from the Gulf of Guinea coastline. They have shown that surface concentrations of pollutants in Savè ($8^o2$'N / $2^o29$'E, Benin) are impacted by emissions from Accra, Cotonou and Lomé and exhibit a marked diurnal cycle with a maximum between 1800 and 2200 UTC associated with pollution transport from Cotonou.

In the following, we analyse the influence of the coastal SST on the diurnal cycle of pollutant transport inland in Savè by comparing WarmES (Figure 9d) and ColdES (Figure 9g) with RefES (Figure 9c, f). It is worth noting that, as pointed out in Deroubaix et al. (2019), only the emissions from Accra, Lomé and Cotonou contribute to the tracer budget in Savè. Furthermore, all three simulations exhibit a maximum in tracer concentration around 2200 UTC originating mainly from Cotonou. Another peak due to Cotonou tracers at around 0500 UTC is present in RefES and WarmES but not in ColdES

(Figure 9c, d, g).

When comparing the diurnal cycles from WarmES and RefES (Figure 9e), it appears that the transport of pollution to Savè is enhanced between 0100 and 1800 UTC when the total concentration of tracers in WarmES is greater than in RefES (see the black solid line in Figure 9e). From 0100 to 1200 UTC there is a clear enhanced contribution from Cotonou and, to a lesser extend Lomé, to the overall transport in Savè. During the rest of the period, it appears that Accra and Lomé contribute to the

excess of tracers concentration, while the contribution from Cotonou diminishes. There is also a significant reduction of the tracer concentration maximum at 2200 UTC which is related to a diminution of the transport from Cotonou. The transport of pollutant is higher for cities that affect Savè at night (namely Cotonou) and in the morning (namely Lomé and Accra).

When comparing the diurnal cycles from ColdES and RefES (Figure 9h), we can see that the transport of pollution to Savè is unchanged between 0700 and 1900 UTC, but significantly reduced between 1900 and 0700 UTC in ColdES compared to

RefES (see the black solid line in Figure 9h). In the latter case, there are less pollutants transported to Savè from Cotonou (which is the closest coastal major city), and no clear modification of the transport from the other cities.

The nighttime peak of pollutant concentration observed between 2100 and 2300 UTC occurs during the "Atlantic inflow" period as shown by (Deroubaix et al., 2019), which end is linked to the end of convection over the land. During this period, the decrease in the transport of tracers to Savè in both experiments, but more marked in ColdES, may be related the maintenance of



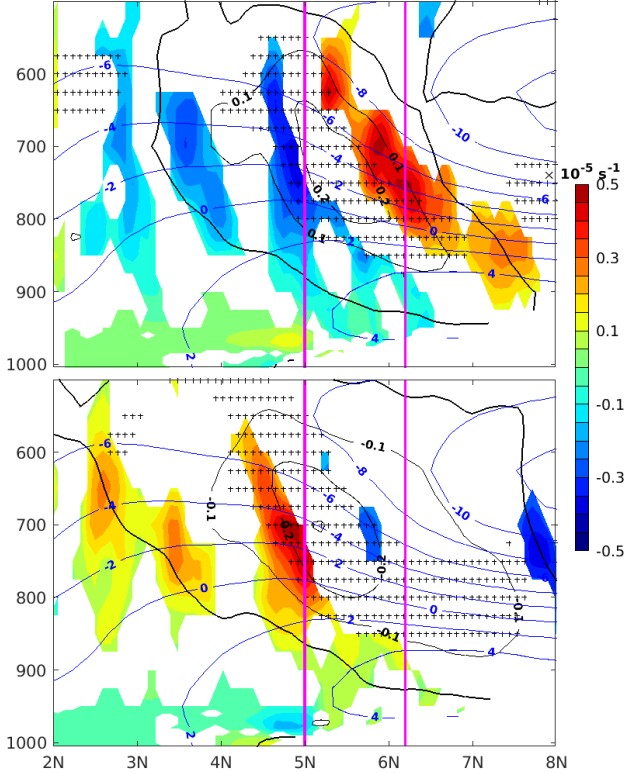

**Figure 10.** June 25th - July 8th, 2016: differences between WarmES and RefES (top) and ColdES and RefES (bottom) within the region between $6^o$W and $2^o$E: zonal velocity (black contours, intervals of 0.1 m.s$^{-1}$, zero contour in heavy line) and relative vorticity (shading). RefES zonal velocity (blue contours, intervals of 2 m.s$^{-1}$). Shading indicates only vorticity anomalies passing a two sampled t-test at the 5% confidence level, while zonal velocity anomalies passing the test are marked with a "+" symbol.

convection over a longer period as a result of warmer coastal SST. In ColdES, the period of reduced transport extends into the "moist morning" period, suggesting that the strengthened coastal upwelling further decreases the intensity of the Low Level Jet. On the contrary, since pollutant transport is enhanced during this period in WarmES, it appears that an anomalously warm upwelling may contribute to strengthen the Low Level Jet in the late night and early morning.

In WarmES, the tracer concentration coming from Cotonou increases by more than 10 % from 0400 UTC to 0800 UTC.
Conversely, in ColdES, tracer concentration coming from Cotonou decreases significantly by less than -5 % from 1900 UTC to 0700 UTC, with the maximum tracer concentration occurring at at 2200 UTC reduced by more than -20 %.

Overall, there is an impact of SST anomalies on the inland transport of pollutants from major cities along the Guinea Coast. Cooler SSTs decrease the inland transport of pollutants, while warmer SSTs increase it. These modifications of the transport occur at night, which suggests an influence on the low-level jet.





## 6 Conclusions


In this study, a set of numerical simulations was used to analyze the seasonal impact of the coastal upwelling that emerges along the Guinea Coast in July. In addition to the reference ensemble, two additional experimental ensembles were produced by decreasing or increasing the strength of the coastal upwelling by about 30%: a stronger (weaker) upwelling leads to a decrease (an increase) in coastal precipitation of about 20-30%, thus explaining why the emergence of the Guinea Coast upwelling

most often coincides with the onset of the little dry season in July. These results confirm the relation between the strength of the upwelling and the humidity transport suggested by Tanguy et al. (2022). The coastal upwelling also has an impact on the distribution of atmospheric pollutants: their northeastern transport in the monsoon flow tends to weaken, reducing their continental concentration, and potentially increasing this concentration on some spots along the coast. The intensity of the upwelling also affects the diurnal cycle of pollutants transport inland, especially during the "moist morning" period when a

weaker upwelling leads to enhanced pollutant transport inland, whereas a stronger upwelling reduces pollutant transport inland.

The mechanisms associated with the coastal upwelling feedback on atmospheric dynamics breaks down as follows: the cooling of the coastal SST induces a slowing of the monsoon flow near the coast; this decreases the coastal convergence of humidity transport, which inhibits deep convection and precipitation. In addition, it decreases the land-sea surface temperature gradient during the night, which weakens the land breeze strength. Since the diurnal alternation of land breeze and sea breeze

forces precipitation to occur in the late night and early morning, a weakening of the land breeze leads to a strong decrease in morning coastal precipitation. This clearly shows that numerical weather models need a good representation of the SST impact on the land and sea breezes in order to produce an accurate forecast of the onset of the little dry season along the Guinea Coast.

Note that our results only demonstrate the impact of the coastal upwelling on the reduction of coastal precipitation south of $6^o$N; they do not indicate any influence further north, although a reduction in precipitation is generally observed during the

little dry season up to $8^o$N (Figure 1). However, it is very interesting to note that the dampening (strengthening) of the coastal upwelling leads to a small deceleration (acceleration) of the AEJ on its southern side between 850 and 600 hPa, just above the coast (Figure 10, black contours), probably resulting from the convective anomalies discussed in section 4. Although this signal is weak, it is nonetheless significant in both experiments and induces significant relative vorticity anomalies (shading). An increase in the latter at the AEJ altitude is known to increase the likelihood of mesoscale convective system formation

and precipitation (see, e.g., Cook 2015), so it seems necessary to further investigate whether a negative vorticity anomaly following the emergence of the coastal upwelling could contribute to the observed reduction in precipitation between $6^o$N and $8^o$N during the little dry season.

Furthermore, the short duration of our simulations of only a few weeks for a given year limits the reach of our findings. Further studies should examine their validity for simulations spanning at least over several years. More detailed simulations

should also provide clearer results over the coastal region east of Cape Three Points: the results were less significant there than further west, maybe because of a smaller amplitude of SST anomaly resulting from the methodology used to dampen the seasonal cooling of the coastal SST.



Finally, the emergence of the Guinea Coast upwelling in July has been shown to oppose the inland monsoon flow. The latter remains a major contributor of humidity to summer precipitation in the Sahel, but generally in early July, it becomes weaker than the western low-level humidity transport from the Tropical North Atlantic (Lele et al. 2015): the timing of this change could then be controlled by the emergence of the Guinea Coast upwelling. Moreover, previous studies have suggested that the westward input of humidity is responsible for most of the interannual and decadal variability in observed summer rainfall in the Sahel (Pu and Cook 2010, Lele et al. 2015), with September being the most variable month. This variability is also probably impacted by global teleconnections, such as the El Nino Southern Oscillation. The coastal upwelling could therefore control the amount of interannual variability from global teleconnections exerted on the summer monsoon in a given year: a better understanding of the interaction between the coastal upwelling and the monsoon flow could help improve the seasonal predictability of the summer monsoon in West Africa.

*Code and data availability.* All data and codes are available on demand.

*Author contributions.* Gaëlle de Coëtlogon and Adrien Deroubaix designed and implemented the numerical simulations and conducted the data analysis. Laurent Menut, Cyrille Flamant and Marco Gaetani provided guidance on the theoretical framework and assisted with the interpretation of the simulation results. All authors contributed to the writing and editing of the manuscript.

*Competing interests.* No competing interests are present.

*Acknowledgements.* This work was granted access to the HPC resources of IDRIS under the allocation 2020 - A0060107454 made by GENCI. The authors want to thank Tatsuo Onishi (LATMOS-IPSL) for his great help in the numerical work. A.D. acknowledge the European Union's Horizon 2020 research and innovation programme for supporting this work under the Marie Skłodowska-Curie grant, agreement No 895803 (MACSECH — H2020-MSCA-IF-2019).



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
