# Peer review of "Impact of the Guinea Coast upwelling on atmospheric dynamics, precipitation and pollutant transport over Southern West Africa"

_EGUsphere, 2023_

## Author Comment (AC1)

**RC1**: 'Comment on egusphere-2023-681', Elsa Mohino, 25 Jul 2023 : **reply by the authors in blue characters**.

**General comments:**

This manuscript analyses the effect of coastal upwelling off the Guinean Coast in coastal precipitation and pollution transport over Souther West Africa in the transition period June – July by means of sensitivity experiments with the regional atmosphere WRF model and the transport CHIMERE model. The simulations consist in a reference experiment using as boundary conditions NOAA SSTs from 2016, and two altered experiments, namely a warm and a cold one, in which upwelling is damped and enhanced, respectively. All experiments provide 10 ensemble members. After analysing the model response, the authors conclude that a stronger upwelling weakens coastal precipitation by promoting less surface wind and moisture convergence locally, especially in the late night/early morning due to the reduction of land breeze. Through modifying winds, upwelling also reduces the inland transport of pollutants produced in major coastal cities, especially at night.

As acknowledged in the manuscript, the main hypothesis on the effect of upwelling on coastal precipitation is not novel. However, the authors provide further evidence on the mechanisms at play and their effect based on the sensitivity experiments performed. They also present novel results related to the effect on pollution transport inland and the conclusions are well supported by their results. I find, however, the manuscript would benefit from additional discussion on the limits of the experimental setup and on the results on pollutant transport (see specific comments section). The scientific approach is clearly outlined, though some clarifications are still needed on the experimental setup (see specific comment section). Formally, I find the manuscript clear, well structured and written, and overall easy to follow, just some minor corrections and clarifications are still needed (see technical corrections). The title and abstract are clear and present the main ideas and results of the manuscript. All in all, I support the publication of the manuscript after modifications following the comments below.

**Specific comments:**

1) In general, coastal upwelling is heavily influenced by surface winds. The modification of surface winds shown in their experiments could well feedback onto the upwelling altering it. However, the authors do not take into account this possible atmosphere-ocean coupling not only in designing the experiments but also on commenting the results and limits of their findings. I suggest to revise the introduction to take ocean-atmosphere coupling into account in the region and add a discussion in the last section on the limits of their experimental setup and possible impact on their interpretation of the results.

First, we want to express our sincere gratitude to Dr. Mohino for her careful review of our study and her constructive and very helpful suggestions. Below, you will find our responses to the questions in blue characters to facilitate distinction.

It is indeed true that our study focuses solely on the influence of sea surface temperature (SST) on the atmospheric circulation and does not address the opposite influence. Therefore, we propose to include the following statement in the introduction:

"Note that the dynamics of coastal upwelling remain poorly understood, but it is likely forced by local or remote surface winds, as demonstrated, for instance, in Djakoure et al. (2017). They conducted idealized numerical experiments and found that while the coastal upwelling west of Cape Three Points is highly sensitive to inertia and Guinea Current detachment from the coast, the upwelling east of Cape Three Points is mainly induced by local winds through the divergence of Ekman transport. This is further supported by the modeling study of Da-Allada et al. (2021), which found that, west of Cape Three Points, a SST cold event in February–June 2012 was largely explained by enhanced vertical mixing caused by the strengthened Guinea Current, whereas east of Cape Three Points, a major contributor to the SST change was the zonal wind. The SST in the Guinea coast upwelling is thus significantly influenced by the Guinea Current and surface winds. The question being studied here is whether it can, in turn, influence atmospheric circulation and precipitation.

Indeed, at interannual timescales, [...]"

Additionally, in the conclusion, we propose to add the following statement:

"Therefore, coastal upwelling could potentially regulate the level of interannual variability resulting from global teleconnections affecting the summer monsoon in a particular year. However, the SST of the coastal upwelling is heavily impacted by surface winds and the Guinea Current (Djakoure et al. 2017): considering that the Guinea Current is also likely influenced by surface winds but on a larger scale, there might be a possibility of better understanding the interaction system between coastal upwelling and monsoon flow. A study with a coupled ocean-atmosphere model is then needed to further investigate the potential influence of this interaction on intraseasonal variability in southern West Africa, which could lead to improved seasonal predictions of the summer monsoon."

2) I wonder to what extent we can trust the results shown on the pollution transport. Are there any observations or different model results to compare with, even for the reference simulation?

Yes, the simulations have been compared to the DACCIWA campaign observations in a prior study (Deroubaix et al. 2022). Therefore, we suggest adding the following text to the article, specifically at line 294 in the previous version of the paper:

« The aerosol simulations conducted using the same WRF configuration as the reference simulation in our study were compared to the DACCIWA campaign observations, as outlined in Deroubaix et al. (2022). The findings from this comparison highlighted that the model effectively reproduces the urban plumes of Accra and Abidjan. This success was achieved through two experiments aimed at modeling the dispersion of anthropogenic aerosol emissions from megacities along the Guinea coast. These experiments also facilitated an investigation into the impact of changes in these emissions on meteorological conditions. The range of simulated aerosol concentrations in the experiments is grounded in realism. Notably, the simulation that exhibits the closest agreement with the observations was intentionally designed to exaggerate aerosol emissions, exceeding measured levels by a factor of 10, as observed a few years prior to 2016. This augmentation of anthropogenic emissions aligns with projections of rapid and 'explosive growth' in emissions, as indicated by Liousse et al. (2014). »

3) The authors do not explain the choice of trend correction used in the warmES and coldES experiments. If I understand correctly, inside the delimited region of -0.052ºC/day linear trend, for the warmES experiment, the linear trend at all grid points is set at -0.052ºC/day. Why? Why not a stronger or weaker trend?

That was indeed a somewhat arbitrary selection. However, it was motivated by the desire to mitigate the cooler sea surface temperature (SST) region near the coastline, as shown in Figure 2. Linear trends were computed and presented in Figure 5. Subsequently, we opted for a trend value that roughly corresponded to the 27°C isotherm in Figure 2. Within the selected coastal area, we ensured that all trend values are equal to -0.052°C/day. This resulted in a warm SST anomaly with a maximum increase of 1°C closest to the coast west of Cape Three Points during the final two weeks of the simulation (Figure 6a). On average, spanning from 6W to 2E, this generated an anomaly slightly exceeding 0.5°C (as depicted in Figure 7d). This approach suited our purpose, as we aimed to represent a significant SST anomaly relative to the seasonal upwelling, which causes a cooling of up to 3°C, while staying within a realistic range compared to interannual variability. Therefore, we maintained this threshold value to constrain the trend within the experimental region.

We propose to replace the following part of the text, l. 144-146 :

« ii) a threshold value of -0.052°C/day is chosen in order to delimit the region where the SST is to be modified (Figure 5, top, black contour). This value fits the 27°C SST contour,

i.e. approximately the edge of the coastal upwelling (Figure 3a). Out of this area, SST timeseries remain unchanged. »

by :

« ii) A threshold value of -0.052°C/day has been deliberately selected to define the boundary of the region in which the SST is to undergo modification (Figure 5, top, delineated by the black contour). This specific value has been chosen in a purposeful manner, approximately aligning with the 27°C SST contour, which corresponds to the edge of the coastal colder zone (Figure 3a). The rationale behind this choice lies in its ability to encompass areas where the average SST value remains below 27°C throughout the final two weeks of simulation. Beyond this demarcated region, the SST time series remain unaffected. »

4) I couldn't find what were the lateral atmospheric boundary conditions used in the experiment. Please clarify.

Apologies for the oversight. We have included the following lines in the text (line 65) to direct interested readers to the detailed simulation setup as outlined in Deroubaix et al. 2019:

'Meteorological initial and boundary conditions are obtained from operational analyses generated by the US National Center for Environmental Prediction (referred to as operational analyses; for a comprehensive description, please consult Deroubaix et al. 2019).

**Technical corrections:**

- Line 42: two instances of "based" in the same sentence. Rephrase to avoid repetition.

Done, the new sentence is : « However, their results were based on composite analyses derived from an empirically determined date of the apparition of coastal upwelling.

- Line 43: why do you use bold here ?

Last minute typo error, sorry !

- Line 72: remove excess of points.

Done.

- Lines 94-96: "In addition, east of ..." I cannot follow this sentence. Could you rephrase, please?

The new sentence is the following: « Furthermore, east of 2°W, the wind intensifies and shifts to an easterly direction along the Togo-Benin coastline. This shift is likely influenced by a positive SST gradient downwind in the northeastward direction, a meridional surface temperature gradient between land and sea, or the coastal topography (Flamant et al., 2018). »

- Lines 104-105: "... because the near-surface ..." The higher content of water vapor on the continent than over ocean is not discussed later on, right? This is a bit confusing to me. Could you explain it more, please?

Sorry for the poor formulation. Here is the new sentence, in a much simpler way:
« because the water vapor content in the lower atmosphere is higher over the land than the ocean, as can be seen in Figure4b. »

- Line 117: "These two convergence areaS .."

Done.

- Line 148: time serie → time series (both instances)

Done.

- Line 149: remove "capped". If I understood correctly, your methodology sets the linear trend in all points inside the modified region to -0.052°C/day.

Correct. The new sentence is « All time series in the upwelling area thus have a linear trend of -0.052°C/day. »

- Lines 153-154: I do not follow how you conclude that the damping effect on warmES is 1/3. If I understood correctly you methodology, in point A and B (an all points inside the modified area) the linear trend is set to -0.052°C/day, so the total change between June 5th to July 7th would be: 34 days times -0.052°C/day = -1.77 °C, which is not 1/3 of 3°C. Please clarify.

The anomaly we are discussing is the difference between the reference and experimental simulations, rather than the extent of SST cooling over the course of the simulation period. To illustrate, at point A, the trend within the reference timeseries stands at approximately -0.09°C/day (trend in RefES). If we subtract -0.052°C/day (trend in WarmES)

from this value, we arrive at -0.038°C/day, which is indeed the underlying tendency responsible for the anomaly.

Furthermore, for the purpose of enhancing the signal-to-noise ratio in our findings, we conduct an averaging process over the period spanning from June 25th to July 7th before conducting a comparative analysis between the experimental and reference simulations. Consequently, we need to consider the resultant anomaly after 27 days (averaging between day 20 and day 34), rather than 34 days. This approach yields an anomaly of -0.038°C/day multiplied by 27 days, resulting in an approximate increase of 1°C (in line with what is depicted in Figure 6a). We believe that a simple swap between the last two sentences in the paragraph should clarify this aspect:

« The resulting WarmES SST anomaly, averaged over the last two weeks of the simulation, exceeds 1°C off Ivory Coast and is approximately 0.7°C off Ghana (Figure 6a, black contours). Because the reference SST decreases by about 3 degrees between June 5th and July 8th at these locations, it means that the magnitude of the coastal upwelling is dampened by about one third in WarmES. »

- Line 158-159: same comment as below regarding the coldES experiment and the 1/3 enhancement.

Same answer applies: the SST anomalies in WarmES and ColdES are completely symmetrical.

- Line 255: Why do you use "arbitrary units" in the concentrations? How can you be sure to add same contributions? Why not use some real unit?

The tracer emissions from the five cities are treated in the model as a passive gas, emitted in the center of the city (point emission). In the model, we define an emission flux of tracers emitted during the period. Each city's tracer emission fluxes are scaled by population. Thus in each box of the grid at each hour of the period, the model simulates a concentration of urban tracers of these five cities. We therefore have the total concentration of urban tracers (the sum of the five concentrations) with the contribution of the five cities (the proportion of the five concentrations).

In practice, we set a ton emitted (continuously) per day during the period for Lome (and we set a higher quantity for the four other cities scaled by population), which in this case corresponds to a concentration of tracers in ppb. However, this is an arbitrary choice because depending on the quantity emitted, the unit could be the ppt (if this quantity is divided by a thousand) or the ppm (if this quantity is multiplied by a thousand).

Using a real unit such as ppb would be confusing to the reader, who could interpret it in terms of quantity of pollution, whereas we want to interpret the relative changes in

concentration (in percentage) due to the change in transport between the WarmES and ColdES experiments.

- Line 265: "… negative anomalies to the southeast …" is this correct?

The negative values are indeed somewhat intertwined with positive values there, so we have included the term "dominant" in the sentence: "… dominant negative anomalies in the southeast."

- Line 275: WarmES → ColdES

Done.

- Line 279: "cover(Schuster" → "cover (Schuster"

Done.

- Line 313: "which end" → "whose end" or "the end of which"

- Line 313: "During this period.." Which period? Two different have just been mentioned (the nighttime peak and its end).

- Line 314: "marked" → "markedly"

- Line 314: "may be related the" → "may be related to the"

- Line 314-315: "may be related …warmer coastal SST." I don't follow this explanation. Coastal SSTs are colder in ColdES, right?

- Line 317-318: How does this possible strengthening of the low level jet in the warmES experiment related to the weaker transport in this simulation. Overall I find this paragraph confusing.

We agree with the reviewer that this part is not clear. Especially since we do not wish to focus on this period because we can consider that the maximum at 2200 UTC is similar for the three simulations. This part has been deleted (between « The nightime peak of pollutant concentration … » and « … as a result of warmer coastal SST. »).

On the other hand, there is a change at night, which corroborates the previous results, and which we now quantify. The different modifications made throughout this section in the paper are indicated at the end of the present document.

- Line 325: Perhaps change the title of the section to "Conclusions and discussion"?

Done.

- Figure 1: increase the size of numbers in colorbars

Done.

- Figure 2: increase size of colorbar and text fonts.

Done.

- Figure 3: add the coast line to the plots.

Done.

- Figure 4: what are the arrows in panels a and b representing? Is it meridional and vertical wind ? Please clarify in the caption and add reference arrow

It is indeed meridional and vertical wind. It was clarified in the caption and vertical scale for arrows was added (horizontal scale is given by shadings).

- Figure 7: improve the quality of panels a and b. It is difficult to see the arrows, contours and labels due to the low resolution of the panels.

Done.

- Figure 8: Is the zonal wind represented in the arrows as the x component? Please add this info into the caption because it is weird to show the zonal wind in a projection of time of the day.

Done, the following sentence has been added to the caption: « The horizontal scale of the vectors represents a velocity, despite being aligned in the direction of time. »

- Figure 9: add the letters to the plots. In plots a and b, add the location of the five major coastal cities. Enhance size of numbers to make them more consistent with other figures.

Done.

- Figure 10: enhance resolution (or better use vector plot). It is horizontal relative vorticity in shading, right? Could you clarify?

Done. Yes, that's correct. It's horizontal relative vorticity: we included it in the caption.

Changes made in section 5 (in red) :

In the following, we analyze the influence of the coastal SST on the diurnal cycle of pollutant transport inland in Savè by comparing WarmES ( Figure 9 d) and ColdES ( Figure 9 g) with RefES ( Figure 9 c, f). It is worth noting that, as pointed out in Deroubaix et al. (2019), only the emissions from Accra, Lomé and Cotonou contribute to the tracer budget in Savè. Furthermore, all three simulations exhibit a maximum in tracer concentration around 2200 UTC originating mainly from

Cotonou with a comparable magnitude, followed by a period of high tracer concentrations originating from Cotonou, Lomé and Accra until . Another peak due to Cotonou tracers at around 0900 UTC, and a decrease in tracer concentration from 0900 to 1800 UTC  is present for the three simulations in RefES and WarmES but not in ColdES (Figure 9c, d, g).

When comparing the diurnal cycles from WarmES and RefES (Figure 9e), it appears that the transport of pollution to Savè is enhanced between 0100 and 1800 UTC when the total concentration of tracers in WarmES is greater than in RefES (see the black solid line in Figure 9e). From 0100 to 1200 UTC, there is a clear enhanced contribution from Cotonou and, to a lesser

extent Lomé, to the overall transport in Savè. During the rest of the period, From 1300 to 1800 UTC, it appears that Accra and Lomé contribute to the excess of tracers concentration, while the contribution from Cotonou diminishes. There is also a significant reduction of the tracer concentration maximum at 2200 UTC which is related to a diminution of the transport from Cotonou. The transport of pollutants is higher for cities that affect Savè at night (namely Cotonou) and in the morning (namely Lomé and Accra).

When comparing the diurnal cycles from ColdES and RefES (Figure 9h), we can see that the transport of pollution to Savè is unchanged between 0700 and 1900 UTC, but significantly reduced between 1900 and 0700 UTC in ColdES compared to RefES (see the black solid line in Figure 9 h). In the latter case, there are less pollutants transported to Savè from Cotonou (which is the closest coastal major city), and no clear modification of the transport from the other cities.

The nighttime peak of pollutant concentration observed between 2100 and 2300 UTC occurs during the "Atlantic inflow" period as shown by (Deroubaix et al., 2019), which end is linked to the end of convection over the land. During this period, the decrease in the transport of tracers to Savè in both experiments, but more marked in ColdES, may be related the maintenance of convection over a longer period as a result of warmer coastal SST.

In ColdES, the period of reduced transport extends into the "moist morning" period, suggesting that the strengthened coastal upwelling further decreases the intensity of the Low Level Jet. On the contrary, since pollutant transport is enhanced during this period in WarmES, it appears that an anomalously warm upwelling may contribute to strengthen the Low Level Jet in the late night and early morning. In WarmES, the tracer concentration coming from Cotonou increases by more than 10 % from 0400 UTC to 0800 UTC. Conversely, in ColdES, tracer concentration coming from Cotonou decreases significantly by less than -5 % from 1900 UTC to 0700 UTC, with the maximum tracer concentration occurring at at 2200 UTC reduced by more than -20 %.

---

## Author Comment (AC2)

**RC2**: 'Comment on egusphere-2023-681', Anonymous Referee #2, 04 Oct 2023 : **reply by the authors in blue characters**.

In this manuscript entitled 'Impact of the Guinea Coast upwelling on atmospheric dynamics, precipitation and pollutant transport over Southern West Africa' de Coëtlogon et al. analyse the mechanism by which a zonal band of precipitation over West Africa is pushed North over the Sahel starting in late June/early July. Their work and the simulations presented are based upon the proposition of Tanguy et al. (2022), based on satellite observations, that the main driver for pushing North the precipitation are the sea-surface temperatures of the coastal upwelling near the coast of Guinea.

In contrast to Tanguy et al. (2022) this work is based on the analysis of two ensemble of simulations, one with warmer SSTs near the Guinean coast and a second one with warmer SSTs than the ensemble that was run as the control. The SST anomaly created amounts to 0.5°C on June 15th and 1.0°C on July 7th, the variability is conserved by the method. The diagnostics presented in the manuscript show how the meridional circulation changes in response to these warmer and colder temperatures, entraining (respectively slowing down) the transport of water vapor and hence the precipitation inland (northward from the Gulf of Guinea). These SST anomalies dampens the precipitation in the case of the cold upwelling by decreasing the convergence of humidity whereas the reverse is true when SSTs are increased (warm upwelling). The analysis through the diagnostics presented in the paper is convincing and the two ensemble of simulations illustrate well the role played by the convergence of humidity in this coastal region.

First, we want to express our sincere gratitude to the anonymous reviewer for their careful review of our study. Below, you will find our responses to the questions in blue characters to facilitate distinction.

I have two remarks that might improve the manuscript.

The introduction focuses solely on how the Sahel precipitation is affected by the mechanism at hand that is the variations of SST near the coast of West Africa and in particular near Guinea and Benin. It would be useful to remind the reader that other processes play a role in the position and the strength of the precipitation over the Sahel as described by the following authors: Haywood et al., 2016; Miller et al., 2014 and Balkanski et al. 2021.

Thanks for this suggestion, the following lines were therefore added in the first paragraph of the introduction : « The location and intensity of Sahel precipitation in boreal summer are mainly controlled by the zonal tropical overturning circulation and the mid-tropospheric African Easterly Jet (AEJ) and high-tropospheric Tropical Easterly Jet (TEJ), which are maintained by two diabatically forced meridional circulations, one associated with deep moist convection and a second from dry convection to the north over the Sahara (Haywood et al. 2016). In addition, aerosols play an important role, as the absorption of sunlight by dust affects not only surface temperature but also surface wind speed, vegetation and humidity transport, leading to complex

interactions (Miller 2014). Balkanski et al. (2016) also highlighted the potential for a stabilizing feedback loop involving dust emission, atmospheric absorption, and Sahel precipitation in climate models. »

The second remark concerns the analysis of the effect on pollution which is analysed for the five main cities affected by the changes in circulation brought about by the upwelling temperature (Abidjan, Ivory Coast; Accra, Ghana; Lom, Togo; Cotonou, Benin and Lagos, Nigeria). Using a generic tracer for pollution gives an information about the relative variations that pollutants will incur but it would be much more informative to have run a model with a full or a simplified chemistry to study how this translates into concentrations for the main pollutants that are: O3, NOx, SOx.

This study is above all based on the study of dynamical processes, and in this case, the use of passive tracers has an advantage: the simulated concentrations accurately reflect the variations in meteorology and only in this process. Active chemistry cannot change the values and thus alter the interpretation linked to the meteorological process studied. The reviewer is right to say that chemical modeling would provide other answers. But the article already seems long enough to us and this would, in our opinion, introduce a 2nd subject in this article. We are keeping this interesting idea in mind and we will probably be able to make another study of it: we will then have to completely revise the chemical boundary conditions (very important in this region with sea-salt, fire and dust emissions and transport), the high-resolution anthropogenic emissions inventory over the region and implement everything in a coupled configuration of the WRF and CHIMERE models. It's a consequent work, and a long way from the present study. We nonetheless changed the title of this section form « Pollutant transport » into « Tracers transport », which seems to us more representative of its content.

All references in the manuscript need to be checked as some are incomplete.

For example:

Tanguy, M., De Coëtlogon, G., and Eymard, L.: Sea surface temperature impact on diurnal cycle and seasonal evolution of the Guinea coastrainfall in Boreal spring and summer, Monthly Weather Review, ?, ?, https://doi.org/?, 2022.

Done.

Taking into consideration the two remarks this manuscript is worthy to be published in Atmospheric Chemistry and Physics.

Again, thanks a lot for the careful reading !

references cited:

Tanguy, M., De Coëtlogon, G., and Eymard, L.: Sea surface temperature impact on diurnal cycle and seasonal evolution of the Guinea coastrainfall in Boreal spring and summer, Monthly Weather Review, 150, 12, pp. 3175-3194, https://doi.org/10.1175/MWR-D-21-0155.1, 2022.

Haywood, J. M., Jones, A., Dunstone, N., Milton, S., Vellinga, M., Bodas-Salcedo, A., Hawcroft, M., Kravitz, B., Cole, J., Watanabe, S., and Stephens, G.: The impact of equilibrating hemispheric albedos on tropical performance in the HadGEM2-ES coupled climate model, Geophys. Res. Lett., 43, 395–403, https://doi.org/10.1002/2015GL066903, 2016.

Miller, R. L., Knippertz, P., Pérez García-Pando, C., Perlwitz, J. P., and Tegen, I.: Impact of Dust Radiative Forcing upon Climate, in: Mineral Dust: A Key Player in the Earth System, edited by: Knippertz, and Stuut, J.-B. W., Springer Netherlands, Dordrecht, 327–357, https://doi.org/10.1007/978-94-017-8978-3_13, 2014.

Balkanski, Y., Bonnet, R., Boucher, O., Checa-Garcia, R., and Servonnat, J.: Better representation of dust can improve climate models with too weak an African monsoon, Atmos. Chem. Phys., 21, 11423–11435, https://doi.org/10.5194/acp-21-11423-2021, 2021.